# Mechanism of Shrinkage in Compacted Graphite Iron and Prediction of Shrinkage Tendency

**DOI:** 10.3390/ma15238413

**Published:** 2022-11-25

**Authors:** Zeyu Liu, Dequan Shi, Guili Gao, Yicheng Feng

**Affiliations:** 1School of Materials Science and Chemical Engineering, Harbin University of Science and Technology, Harbin 150040, China; 2School of Materials and Chemistry, University of Shanghai for Science and Technology, Shanghai 200093, China; 3School of Mechanical and Energy Engineering, Shanghai Technical Institute of Electronics and Information, Shanghai 201411, China

**Keywords:** compacted graphite iron, shrinkage, mechanism analysis, thermal analysis, artificial neural network

## Abstract

Shrinkage greatly influences the mechanical and fatigue properties of compacted graphite iron and it is necessary in order to study the causes of shrinkage in compacted graphite iron and to predict it effectively. In this paper, a kind of cylindrical necking test sample was designed to evaluate the shrinkage in compacted graphite iron, and a method to calculate the size of shrinkage was proposed. By observing the microstructure around the shrinkage zone, it is concluded that concentrated shrinkage mainly appears in the solidification region where the dendritic gap is closed, and the isolated shrinkage mainly occurs in the final solidification region, and the supersaturated carbon elements are gathered on the surface of the shrinkage. The cause of shrinkage in compacted graphite iron is caused by its solidification method, where the austenite dendrites and the eutectic clusters are generated close to the melt zone during the solidification process, leading to the inability to feed the shrinkage. Based on the thermodynamic analysis, the equations between the volume change of each phase, solid phase rate, and time during solidification of compacted graphite iron were established to theoretically explain the formation mechanism of the shrinkage. Taking nine parameters such as the chemical elements and characteristic values of thermal analysis as the input nods, a four-layer BP neural network model for predicting the size of shrinkage in compacted graphite iron was constructed, and the R-squared of the model reached 97%, which indicates it could be used to predict the shrinkage tendency.

## 1. Introduction

Compacted graphite iron has excellent thermal conductivity and anti-vibration and anti-fatigue properties, and can be used for a longer period of time under the complex working environment of diesel engines, and, at the same time, it has good casting properties and mechanical properties. This is inseparable from the different roles played by the various forms of carbon in the material. In addition, these excellent properties provide the material basis for the development of high-performance diesel engines [1,2,3,4,5,6].

Shrinkage is a key factor affecting the performance of compacted graphite iron castings, and it can lead to a serious decline in the performance of castings. In recent years, research on the shrinkage of compacted graphite iron has been focused on and developed [7,8]. Vazehrad et al. [9] performed a test specimen by cutting from a diesel engine cylinder block in the region of different cooling rates, and found that the shrinkage tendency of the castings with different cooling rates was different and it was also related to the nodularity. Lekakh et al. [10] designed a set of cylindrical specimens arranged in a ring to express the shrinkage of the spheroidal graphite iron by simulation and actual pouring, performed the analysis of the solidification process and modeled the nodularity of graphite spheres as a mathematical function of the shrinkage rate. Boeri et al. [11] designed three different shapes of stepped specimens to evaluate the compacted graphite iron, and it was found that the shrinkage was mainly generated at the interstices of the dendritic arms. Ramos et al. [12] observed the microstructure of compacted graphite iron near the shrinkage zone and found that the eutectic undercooling plays a key role in the microstructure of compacted graphite iron, as well as in the generation of shrinkage, and it was found that a decrease in the eutectic undercooling could reduce the generation of shrinkage.

On-line inspection of the solidification process using the thermal analysis technique is currently a rapid method for the inspection of cast iron [13]. Kanno et al. [14,15] used the thermal analysis method to predict the shrinkage tendency of spheroidal graphite iron, and a mathematical fitting analysis between the thermal analysis characteristic values and shrinkage tendency was performed to obtain the relationship index for predicting the shrinkage. The experimental results of Fourlakidis [16] showed that the carbonization of hypoeutectic ductile iron could reduce the occurrence of shrinkage, and the formation and growth of shrinkage correlated well with the typical characteristic values of the thermal analysis.

However, the causes of shrinkage in compacted graphite iron were not well described in the above study. In this paper, by simulation and experimental verification, a cylindrical necking test sample was designed to express the shrinkage tendency in compacted graphite iron. Then, the microstructure around the shrinkage zone was observed using a hot alkali corrosion method in order to analyze the causes and growth patterns of shrinkage, and the generation of each phase during solidification was calculated according to the thermodynamic and solidification processes. Finally, the shrinkage tendency of compacted graphite iron was predicted by a four-layer BP (Back Propagation) artificial neural network according to the thermal analysis characteristic values and chemical compositions.

## 2. Experiments

### 2.1. Experiment Materials

An ABP20t medium frequency induction furnace was used to melt the compacted graphite iron, and the cored-wire injection method was used for vermicularization and inoculation. The chemical compositions of molten iron were measured using an optical emission spectrometer (SPECTROLAB S, Kleve, Germany) and carbon-sulfur analyzer (LECO, Saint Joseph, MI, USA). The nominal chemical compositions of compacted graphite iron are shown in Table 1. The test materials with a low sulfur content were used to ensure the quality of the compacted graphite iron melt and the adjustment of the molten iron composition through carburant, ferrosilicon, and ferromanganese.

### 2.2. Design of Shrinkage Test Block

In order to characterize the shrinkage tendency in the solidification process of compacted graphite iron, three kinds of cylindrical necking test samples were designed according to the casting solidification modulus method, and their sizes were Ø26 × 55 mm, Ø30 × 60 mm, and Ø40 × 70 mm. In reference to the shrinkage modulus (about 0.6) of the actual castings, as shown in Figure 1, the moduli of the test samples were set as 0.5, 0.6, and 0.78, respectively.

After pouring, three kinds of samples were cut along the middle symmetry plane, as shown in Figure 2. It can be found that with the increase in the sample diameter and casting modulus, the appearance of hole-like defects gradually increased, and the inner surface of defects gradually presented as being smooth and round, which shows the characteristics of the gas hole. Therefore, the cylindrical necking test sample with a large modulus was more likely to form a gas hole. The size of the gas holes was larger than the shrinkage, which would result in a large shrinkage rate for the evaluation. Consequently, the cylindrical necking test sample with Ø26 × 55 mm was more suitable.

In order to further reduce the effect of the gas hole on the shrinkage evaluation, the neck size of the sample was further optimized by changing its height and depth. Four samples with different neck heights and depths were designed, and the neck sizes are shown in Table 2. The numerical simulations were used to appraise the solidification of the molten iron in the samples, and the results are shown in Figure 3. It can be seen that the neck depth and height had a decisive effect on the final solidification of the molten iron. The greater the neck depth and height, the lower the final solidification area, and the slenderer it also becomes. Figure 4 shows the possible zones of shrinkage in the four samples. As the final solidification area moved downward, the possible zones of shrinkage also moved downward, and the increase in neck depth and height led to the shrinkage becoming larger and more concentrated. Finally, the shrinkage was gradually separated into two parts.

The results of the actual pouring castings using the above samples are shown in Figure 5. The shrinkage became more obvious as the neck depth and height increased, but the gas in an area with too low solidification could not be discharged from the upper of the molten iron, resulting in a significant gas hole. Therefore, considering the effect of the gas hole on the shrinkage, the final cylindrical necking test sample is shown in Figure 6. The height and depth of neck were 5 mm and 2 mm, and the distance from the top was 20 mm.

When the shrinkage of compacted graphite iron was evaluated using the designed cylindrical necking test sample, the volume of the shrinkage could not be calculated directly because most of the shrinkage being small and scattered. Therefore, the area around the shrinkage zone could be measured, and the shrinkage intensity index was used to evaluate the shrinkage zone with the same area. The shrinkage was quantified from two scales of the shrinkage area and the shrinkage intensity index, and the statistical method was as follows [17].

(1)Measure the total area of the shrinkage zone on profile *A*_s_. A rectangle was used to mark the shrinkage zone, and the rectangular area was be as small as possible under the premise of including all of the defects in the shrinkage zone. The area of the rectangle was marked as *A*_S_.(2)The shrinkage intensity index (*I*_S_) was defined according to the shrinkage diameter. When the shrinkage diameter did not exceed 0.5 mm, it was defined as micro-shrinkage, and the shrinkage intensity index *I*_S_ was 1. When the diameter was 0.5 mm to 2 mm, it was defined as macro-shrinkage, and the shrinkage intensity index *I*_S_ was 2.(3)The shrinkage value *P*_i_ was the product of the shrinkage area *A*_S_ multiplied by the shrinkage intensity index *I*_S_, i.e., *P*_i_ = *A*_S_ × *I*_S_.

When the cylindrical necking test sample was poured, a thermal analyzer was used to collect the temperature of the molten iron. The typical cooling curve and thermal analysis characteristic values are shown in Figure 7, and the meanings of the characteristic value points are shown in Table 3. Only the basic characteristic value points are listed here, and more characteristic value points to be used in the model will be introduced later.

## 3. Results and Analysis

### 3.1. Mechanism of Shrinkage Generation in Compacted Graphite Iron

The microstructure around the shrinkage after corroding for 10 s using 4% nitric acid alcohol is shown in Figure 8. It can be seen that the shrinkage was located in the center of the eutectic cluster, and it had an irregular shape and bumpy appearance at the edge. There was light-colored austenite crossing the shrinkage, and there was no graphite inside the austenite.

The shrinkage in the compacted graphite iron was mainly caused by its solidification characteristics and solidification mode. The solidification characteristics of the compacted graphite iron were between spherical graphite iron and flaky graphite iron, and its inside solidification was similar to the spherical graphite iron, showing paste solidification. So, it had a long eutectic solidification time, and there was a large difference between the morphology of the first solidified part and the later one. When the chemical composition was near the eutectic point, there was a solidification pattern similar to that of flaky graphite iron, that is, the precipitation of austenite dendrites followed by graphite generation [18].

The chemical composition of the experimental compacted graphite iron was micro-sub-eutectic, showing first the precipitation of austenite and then vermicular graphite and the austenite eutectic solidification mode. When the eutectic clusters of graphite and austenite were in contact with each other, the unsolidified molten iron was divided into several discontinuous isolated melt pools by austenite dendrites and eutectic clusters, and there was no feeding channel to make up for the shrinkage.

At the same time, with the growth process of graphite eutectic groups, it became larger and larger. When they were in contact with each other, the expansion force caused by the graphite eutectic group made the austenite dendrite gap increase, so that the final solidification part could not be fed and thus the shrinkage formed.

The hot alkali corrosion method was performed to further observed the microstructure and analyze the mechanism of shrinkage formation in compacted graphite cast iron. The parameters of the hot alkali corrosion solution are shown in Table 4, and Figure 9 shows the color metallographic photographs of the shrinkage zone.

It can be seen that there were two locations where shrinkage occurred. It appeared in the dendritic gap, such as in Figure 9a,b. It also appeared in the final solidification zone, as shown in Figure 9c,d.

The solidification around the shrinkage in the dendrite gap was earlier. During the growth of the primary austenite dendrites, they overlapped each other to form a skeleton, and the liquid metal was divided into different zones. After the eutectic growth of the liquid metal in the gap, the dendrite gap also expanded, resulting in shrinkage formation in the dendrite gap. In addition, the cooling time of Figure 9b was later than that of Figure 9a. The austenite around the defects in Figure 9a is light yellow and orange-red, which was the first precipitated austenite dendrite. However, the austenite in Figure 9b is blue, which was the austenite generated by eutectic solidification, and the solidification time was later.

Another kind of shrinkage appeared in the final solidification zone outside the eutectic cluster. The color metallography showed that the zone around the shrinkage was brown, which indicates that the solidification time was the latest. There was no liquid metal to feed this zone, and the graphite was smaller and the eutectic cluster did not grow. Therefore, it could not achieve the self-feeding of graphite, resulting in the formation of shrinkage.

SEM of the interdendritic shrinkage was performed and the results are shown in Figure 10. The interconnected dendrites and the morphology of the interdendritic shrinkage could be clearly seen. The shrinkage was concentrated in the interstices of dendrites, and the dendrites could be found to penetrate through the shrinkage. There was less graphite around the shrinkage, and the graphite size was also smaller. One way to prevent the generation of shrinkage in compacted graphite iron is to control the solidification state of molten iron to be eutectic, which can reduce the amount of austenite dendrites, so as to depress the overlap of austenite dendrites and the formation of enclosed zones.

The EDS analysis of the shrinkage zone is shown in Figure 11. The results show that the inner surface of shrinkage had mainly C without Fe, which indicates that the remaining C in the molten iron finally precipitated on the surface of the shrinkage during solidification. The results of the line scan show that the inner surface of shrinkage had almost C, and there was a certain deviation of Si around the shrinkage. The closer the shrinkage, the less the Si, which indicated that the concentration of Si in the molten iron decreased when the isolated shrinkage was formed. According to the anti-deviation behavior of Si in austenite, the zone around the shrinkage was in the final solidification, and the supersaturated C in the final solidification zone precipitated on the inner surface of the shrinkage in the form of graphite. Therefore, another feasible way to reduce shrinkage was to make the supersaturated C in the molten iron precipitate in the form of graphite, which could have a self-feeding effect due to the low density of graphite.

According to the above analysis, the shrinkage of compacted graphite iron was related to the volume of austenite dendrites and eutectic graphite during solidification. Through the thermodynamic analysis of this process, it can be obtained.
(1)$$\Phi_{\mathrm{All}}=\Phi_{C}+L_{\mathrm{All}}$$
where $\Phi_{\mathrm{All}}$ is the total heat change of the system, $\Phi_{C}$ is the cooling without transformation, and $L_{\mathrm{All}}$ is the release of the total latent heat of crystallization in solidification. 

The cooling of the system without transformation $\Phi_{C}$ could be calculated according to the thermal analysis zero curve $T_{0}$ based on the Newtonian method [19], and the zero curve could be given by Stefanescu [20].
(2)$$\Phi_{C}=c_{L}\times \rho_{L}\times V_{L0}\times \Delta T_{0}$$
where $c_{L}$ is the specific heat of the liquid phase, $\rho_{L}$ is the density of the liquid phase, $V_{L0}$ is the volume of the sample cup, and $\Delta T_{0}$ is the temperature change of the zero curve.

The release of the latent heat of crystallization $L_{\mathrm{All}}$ can be divided into two parts of primary and eutectic latent heat, and the eutectic latent heat can be divided into eutectic austenite latent heat and eutectic graphite latent heat. The release of latent heat of crystallization is calculated as follows:(3)$$L_{\mathrm{All}}=L_{\mathrm{VG}}\times V_{G}+L_{V\gamma }\times V_{\gamma }$$
where $L_{\mathrm{VG}}$ and $L_{V\gamma }$ are the latent heat of the solidification per unit volume of graphite and austenite, and $V_{G}$ and $V_{\gamma }$ are the volume of graphite and austenite, respectively.

The total heat change of the system $\Phi_{\mathrm{All}}$ can be expressed as follows:(4)$$\Phi_{\mathrm{All}}=c_{\mathrm{mix}}\times m\times \Delta T$$
where $c_{\mathrm{mix}}$ is specific heat capacity of mixture, m is the mass of mixture, and ΔT is the temperature change or the undercooling.

The variation of the specific heat capacity of the mixture at a primary crystallization stage $c_{\mathrm{mixP}}$ with time can be expressed as follows:(5)$$c_{\mathrm{mixP}}\left(t\right)=c_{L}\times \varphi_{\mathrm{mL}}\left(t\right)+c_{\gamma }\times \varphi_{m\gamma }\left(t\right)+c_{G}\times \varphi_{\mathrm{mG}}\left(t\right)\;$$
where $c_{\mathrm{mixP}}$ is the specific heat of the mixture at the primary crystallization stage, $\varphi_{\mathrm{mL}},\varphi_{m\gamma },\varphi_{\mathrm{mG}}$ are the mass fraction of the liquid phase, austenite phase, and graphite phase, respectively, and all of the $c_{\mathrm{mixP}},\varphi_{\mathrm{mL}},\varphi_{m\gamma },\varphi_{\mathrm{mG}}$ vary with time t.
(6)$$c_{\mathrm{mixP}}\times \rho_{L}\times V_{L0}\times (T_{L}-T_{t})=\Phi_{C}+L_{V\gamma }\times V_{\gamma P}$$
where $T_{L}$ is the temperatures of the liquid phase line, $T_{t}$ is the temperatures at time of t, and $V_{\gamma P}$ is the volume of austenite in the primary crystallization stage. 

The change in the solid phase mass is mainly related to the nucleation of primary austenite, and the main driving force of nucleation is undercooling. The greater the undercooling, the greater the driving force, and the bigger the reaction rate. According to the equation of the metal nucleation rate, it can be expressed as [21]:(7)$$I=K\times e^{-\frac{\Delta G_{A}}{kT}}\times e^{-\frac{16}{3}\frac{\pi \sigma_{LC}^{3}T_{0}}{L^{2}\Delta T^{2}kT}}$$
where I is the metal nucleation rate, $\Delta G_{A}$ is the diffusion activation energy of liquid metal atoms crossing the solid–liquid interface, K is nucleation coefficient (constant), k is Boltzmann constant, $\sigma_{\mathrm{LC}}$ is interfacial free energy per unit between liquid phase - nucleus, L is latent heat of crystallization, ΔT is the undercooling, T is temperature, and $T_{0}$ is the equilibrium crystallization point of the metals.

During the process of austenite nucleation, the change in $\Delta G_{A}$ with the temperature is very little, and it is mainly affected by the undercooling degree. The greater the undercooling degree, the greater the nucleation rate. At the pre-nucleating stage, the change in nucleation rate with undercooling can be regarded as linear. Because the composition is almost unchanged, it can be assumed that the solid phase quality is only related to the undercooling, and at the primary crystallization stage it is only determined by the mass of primary austenite. Therefore,
(8)$$\varphi_{m\gamma P}\left(t\right)=k_{1}\cdot \int_{t_{T}{}_{{}_{L}}}^{t}\Delta Tdt$$
where $t_{T}{}_{{}_{L}}$ is the time corresponding to $T_{L}$, $k_{1}$ is the nucleation coefficient of primary austenite, and ΔT is the undercooling, and undercooling is the value on the experimentally determined temperature–time curve corresponding to time *t*. According to the Equations (5) and (8), it can be obtained.
(9)$$c_{\mathrm{mixP}}\left(t\right)=c_{L}\cdot \left(1-k_{1}\cdot \int_{t_{T}{}_{{}_{L}}}^{t}\Delta Tdt\right)+c_{\gamma }\cdot k_{1}\cdot \int_{t_{T}{}_{{}_{L}}}^{t}\Delta Tdt$$
where $c_{\gamma }$ is the specific heat of the austenite.

According to Equations (2), (3), and (9), the following equation can be obtained.
(10)$$V_{\gamma P}\left(t\right)=\frac{\rho_{L}\cdot V_{L0}\cdot \left\{\left[c_{L}\cdot \left(1-k_{1}\cdot \int_{t_{T}{}_{{}_{L}}}^{t}\Delta Tdt\right)+c_{\gamma }\cdot k_{1}\cdot \int_{t_{T}{}_{{}_{L}}}^{t}\Delta Tdt\right]\cdot (T_{L}-T_{t})-c_{L}\Delta T_{0}\right\}}{L_{V\gamma }}$$
where $t\in \left(t_{T_{L}},t_{T_{\mathrm{SEF}}}\right)$.

For the eutectic stage, the heat change of the system is as follows.
(11)$$\Phi_{E}=\Phi_{\mathrm{CE}}+L_{\mathrm{VG}}\cdot V_{G}+L_{V\gamma E}\cdot V_{\gamma E}$$
(12)$$\Phi_{\mathrm{CE}}=c_{L}\times \rho_{L}\times V_{L0}\times \Delta T_{0}$$
where $\Phi_{E}$ is the total heat change at the eutectic stage, $\Phi_{\mathrm{CE}}$ is the cooling without the phase change at the eutectic stage, and $V_{\gamma E}$ is the volume of austenite at the eutectic stage.

Under the premise of constant mass, the following assumptions can be made.

Assumption (1): Only graphite is formed at the eutectic stage, and so there is:(13)$$\rho_{L}\times V_{L0}=\rho_{\gamma P}\times V_{\gamma P}+\rho_{G}\times V_{G}+\rho_{L}\times V_{L}$$

Substituting Equation (13) into Equation (11), the following equation can be obtained.
(14)$$\Phi_{\mathrm{GE}}=\Phi_{\mathrm{CE}}+\frac{L_{\mathrm{VG}}\cdot \left(\rho_{L}\cdot V_{L0}-\rho_{\gamma P}\cdot V_{\gamma P}-\rho_{L}\cdot V_{L}\right)}{\rho_{G}}$$
where $\Phi_{\mathrm{GE}}$ is the heat change of the system when only graphite is formed during solidification.

Assumption (2): Only austenite is formed at the eutectic stage, and so there is:(15)$$\rho_{L}\times V_{L0}=\rho_{\gamma }\times V_{\gamma P}+\rho_{\gamma }\times V_{\gamma E}+\rho_{L}\times V_{L}$$

Substituting Equation (15) into Equation (11), we can obtain the following:(16)$$\Phi_{\gamma E}=\Phi_{\mathrm{CE}}+\frac{L_{V\gamma }\cdot \left(\rho_{L}\cdot V_{L0}-\rho_{\gamma }\cdot V_{\gamma P}-\rho_{L}\cdot V_{L}\right)}{\rho_{\gamma }}$$
where $\Phi_{\gamma E}$ is the heat change of the system when only austenite is formed during solidification.

The actual volume ratio $\varphi_{G}$ of austenite and graphite can be calculated according to Equation (17).
(17)$$\varphi_{G}=\frac{\Phi_{E}-\Phi_{\gamma E}}{\Phi_{\mathrm{GE}}-\Phi_{\gamma E}}$$

Substituting Equations (11), (12), (14), and (16) into Equation (17), the following equation can be obtained.
(18)$$\varphi_{G}=\frac{L_{G}+\frac{L_{\gamma }\cdot \rho_{G}}{\rho_{\gamma }}}{\frac{L_{G}}{\rho_{G}}\cdot a-\frac{L_{\gamma }}{\rho_{\gamma }}\cdot a}\cdot V_{G}$$
where $a=\left(\rho_{L}\times V_{L0}-\rho_{\gamma }\times V_{\gamma P}-\rho_{L}\times V_{L}\right)$, and $V_{L}$, $V_{\gamma E}$, $V_{G}$ are the volumes, which vary with time.

The eutectic growth rate can be described as:(19)$$R=\mu_{1}\cdot \Delta T^{n}$$
where *R* is the eutectic growth rate, ΔT is the undercooling, and $\mu_{1}$ and n are the coefficients of eutectic growth. For the eutectic solidification process of cast iron, n = 2. So, the eutectic growth rate *R* of compacted graphite iron is proportional to the square of the undercooling ΔT [22]. Therefore, the variation of the solid phase mass with time during the eutectic period can be written as:(20)$$\varphi_{\mathrm{mE}}\left(t\right)=\varphi_{m\gamma }+\mu_{1}\cdot \int_{t_{T}{}_{{}_{\mathrm{SEF}}}}^{t}\Delta T^{2}dt$$
where $\varphi_{\mathrm{mE}}$ is the ratio of solid phase mass at the eutectic stage. When $t=t_{T}{}_{{}_{\mathrm{ES}}}$, $\varphi_{\mathrm{mE}}\left(t\right)=1$, and $\mu_{1}$ can be calculated. In addition, the mass ratio of graphite at the eutectic stage can be calculated using Equation (21).
(21)$$\varphi_{\mathrm{mG}}=\frac{\rho_{G}\cdot V_{G}}{\rho_{L}\cdot V_{L}+\rho_{\gamma }\cdot V_{\gamma P}+\rho_{\gamma }\cdot V_{\gamma E}+\rho_{G}\cdot V_{G}}$$
where $\varphi_{\mathrm{mG}}$ is the mass ratio of graphite at the eutectic stage.

The specific heat capacity of the eutectic stage $c_{\mathrm{mixE}}$ is:(22)$$\begin{array}{l} c_{\mathrm{mixE}}\left(t\right)=c_{L}\cdot \left(1-\varphi_{\mathrm{mE}}\left(t\right)\right)+c_{\gamma }\cdot \left[\varphi_{m\gamma P}+\left(1-\varphi_{\mathrm{mG}}\left(t\right)\right)\cdot \left(\varphi_{\mathrm{mE}}\left(t\right)-\varphi_{m\gamma P}\right)\right]\\ \;\;\;\;+c_{G}\cdot \varphi_{\mathrm{mG}}\left(t\right)\cdot \left(\varphi_{\mathrm{mE}}\left(t\right)-\varphi_{m\gamma P}\right) \end{array}$$

According to the law of conservation of mass, the following can be obtained:(23)$$\rho_{L}\times V_{L0}=\rho_{\gamma P}\times V_{\gamma P}+\rho_{\gamma E}\times V_{\gamma E}+\rho_{G}\times V_{G}+\rho_{L}\times V_{L}$$

Equations (11), (12), (22), and (23), give us the following.
(24)$$V_{G}\left(t\right)=\frac{\Phi_{E}-\Phi_{\mathrm{CE}}-L_{\gamma }\cdot \frac{\rho_{L}\cdot V_{L0}-\rho_{\gamma }\cdot V_{\gamma P}-\rho_{L}\cdot V_{L}}{\rho_{\gamma }}}{L_{G}-\frac{L_{\gamma }\cdot \rho_{G}}{\rho_{\gamma }}}$$
where $V_{L}$, $V_{\gamma E}$, $V_{G}$ are volume variables with time.

The values of the thermophysical parameters in the above equations are shown in Table 5. Using Equations (18), (23), and (24), we can obtain $V_{L}$ and $V_{\gamma E}$. According to the thermal analysis curve and the thermophysical parameters in Table 5 [23,24,25], the relationship curves between the phase volume change, solid phase rate of the solidification process and time can be obtained, as shown in Figure 12.

As can be seen from Figure 12a, the total volume of the melt at the primary crystallization stage decreases with the precipitation of the primary austenite, expressing a tendency to shrink. When the precipitation of the primary austenite is increased, the shrinkage of the melt is also accelerated. At the eutectic stage, the precipitation of graphite causes the total volume to expand continuously, which plays a role of self-feeding. The calculated results of the solidification process are in agreement with the previous microstructure observation. The overall curve of the solidification process is shown in Figure 12b. The solid phase rate at the primary crystallization stage increases with the increase in undercooling. The change in solid phase rate at the eutectic stage is the same as the change of the temperature curve. The faster the temperature rises back, the faster the solid phase rate increases. The faster the latent heat of the crystallization releases at the eutectic stage, the faster the temperature rises back. This is consistent with the theoretical analysis. Thus, the solidification of compacted graphite iron can be judged by the volume change of each phase during solidification, and the volume change of each phase can be characterized by the cooling curve of the solidification process.

### 3.2. Shrinkage Prediction in Compacted Graphite Iron

According to formation mechanism of shrinkage, by analyzing the relationship between the thermal analysis curve and the solidification process, some new characteristic points can be achieved, mainly including the characteristic temperature difference of the thermal analysis, the time difference corresponding to the characteristic temperatures, the initial heat release, and the overall heat release of eutectic process. Their meanings and calculation formulas are shown in Table 6, where f(t) represents the temperature–time function. 

A typical correlation analysis was carried out on the parameters, such as the main elements of iron, the characteristic values of thermal analysis, the initial heat release, and the overall heat release, in order to find the correlation between each parameter and the shrinkage, and the results are shown in Table 7 [26]. In this paper, we used the typical correlation analysis function of SPSS software for typical correlation analysis of each group of data, to obtain the data shown in Table 7.

From the canonical loadings, there are nine parameters that have a greater impact on the shrinkage, and they are Mn, P, S, Cu, $t_{1\left(T_{\mathrm{AL}}-T_{\mathrm{EU}}\right)}$, $t_{3\left(T_{\mathrm{ER}}-T_{\mathrm{ES}}\right)}$, $t_{4\left(T_{\mathrm{SEF}}-T_{\mathrm{EM}}\right)}$, $\;Q_{S}$, and $Q_{Z}$. Among them, the parameters of Cu, $t_{1\left(T_{\mathrm{AL}}-T_{\mathrm{EU}}\right)}$, $t_{3\left(T_{\mathrm{ER}}-T_{\mathrm{ES}}\right)}$ and $\;Q_{S}$ are negatively correlated and the rest are positively correlated.

Based on the results of the typical correlation analysis, the artificial neural network with superior nonlinear mapping capability is used to predict the shrinkage tendency. A BP neural network model is established as shown in Figure 13, which includes a 9-node input layer, two hidden layers with 12 nodes and 4 nodes, and a 1-node output layer. The nodes of the input layer are nine parameters obtained from the typical correlation analysis, and the node of the output layer is the shrinkage tendency of iron. After training the model, the results are shown in Figure 14, and its R value is obtained as 97%.

Figure 15 shows the relationship between the shrinkage tendency *P*_i_ and the shrinkage shape. Here the shrinkage is circled in red. The larger *P*_i_, the greater the shrinkage tendency. Therefore, *P*_i_ calculated by the BP model can be used to make an effective prediction of the shrinkage tendency of compacted graphite iron.

## 4. Conclusions

(1)A cylindrical necking test sample was designed to evaluate the shrinkage tendency of compacted graphite iron, which significantly reduces the influence of the gas hole on the shrinkage during solidification. The size of the sample is Ø26 × 55 mm, the height and depth of the neck are 5 mm and 2 mm, respectively, and the neck distance from the top of the sample is 20 mm.(2)Concentrated shrinkage in compacted graphite iron mainly occurs in the solidification zone where the dendritic gap is enclosed. Because this zone cannot be fed, shrinkage forms in the melt solidification process. Isolated shrinkage is mainly formed in the final solidification zone where less graphite formation during solidification makes self-feeding difficult. The supersaturated C gathers on the surface of the shrinkage. The cause of shrinkage in compacted graphite iron is caused by its solidification method, where the austenite dendrites and the eutectic clusters are generated close to the melt zone during the solidification process, leading to the inability to fill the shrinkage.(3)Through thermodynamic analysis, the equations between the volume change of each phase, solid phase rate, and time during solidification were established, and the curves of the volume change of each step and solid phase rate versus time were obtained, though which the formation mechanism of shrinkage was theoretically clarified.(4)Taking nine parameters, such as the chemical elements of iron and the characteristic values of thermal analysis, as the input and the shrinkage as output, a four-layer BP neural network model was constructed to predict the shrinkage. The R-squared of the model reaches 97%, and it can effectively predict the shrinkage tendency of compacted graphite iron.

## Figures and Tables

**Figure 1 materials-15-08413-f001:**
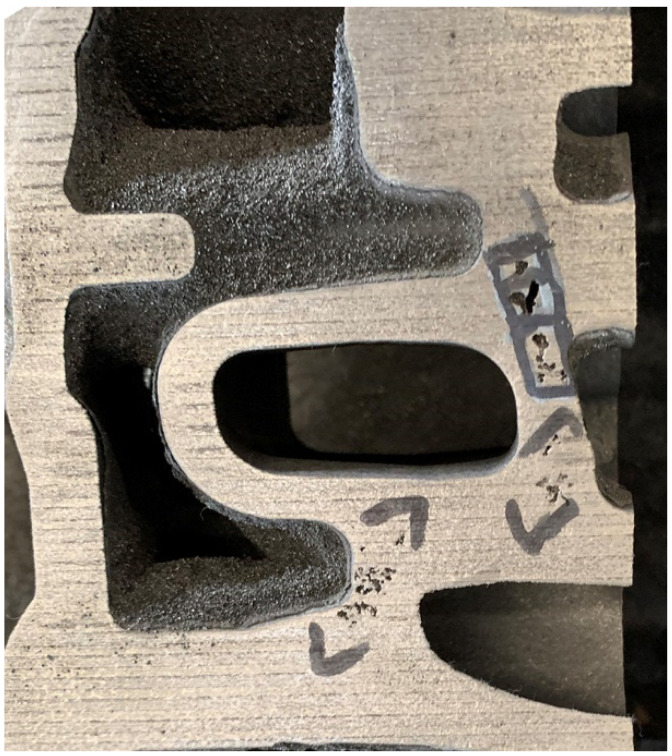
Shrinkage defects in the actual casting with modulus of 0.6.

**Figure 2 materials-15-08413-f002:**
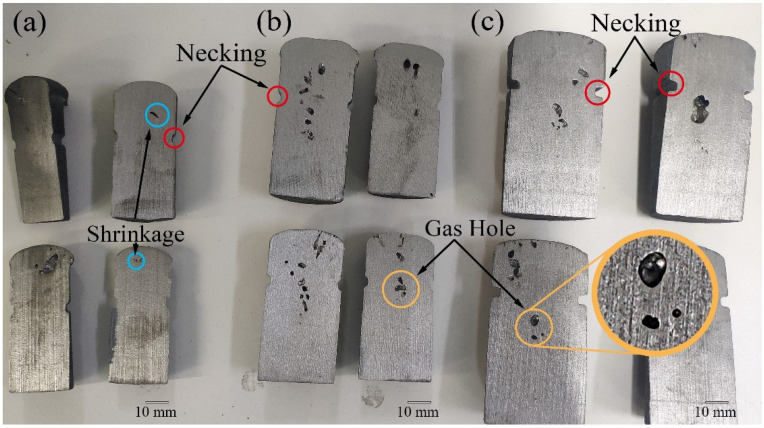
Pouring results of three different sizes of cylindrical necking test samples: (**a**) Ø26 × 55 mm, (**b**) Ø30 × 60 mm, and (**c**) Ø40 × 70 mm.

**Figure 3 materials-15-08413-f003:**
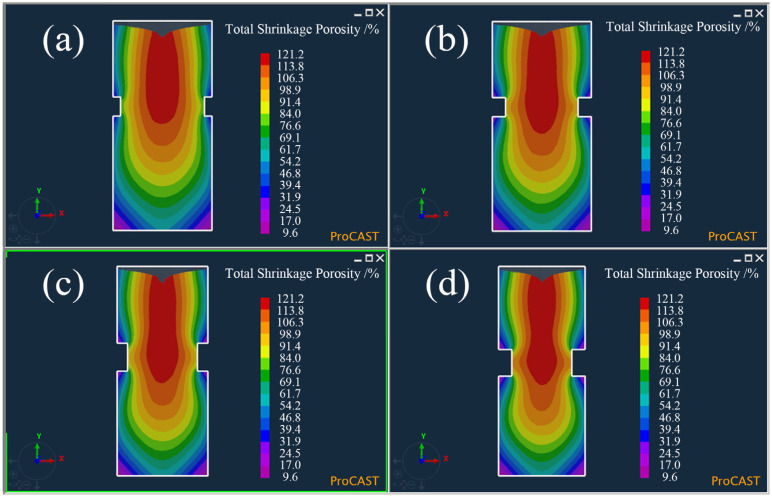
Solidification time of the Ø26 × 55 mm sample with four different neck sizes by simulation: (**a**) No. 1 in Table 3, (**b**) No. 2 in Table 3, (**c**) No. 3 in Table 3, and (**d**) No. 4 in Table 3.

**Figure 4 materials-15-08413-f004:**
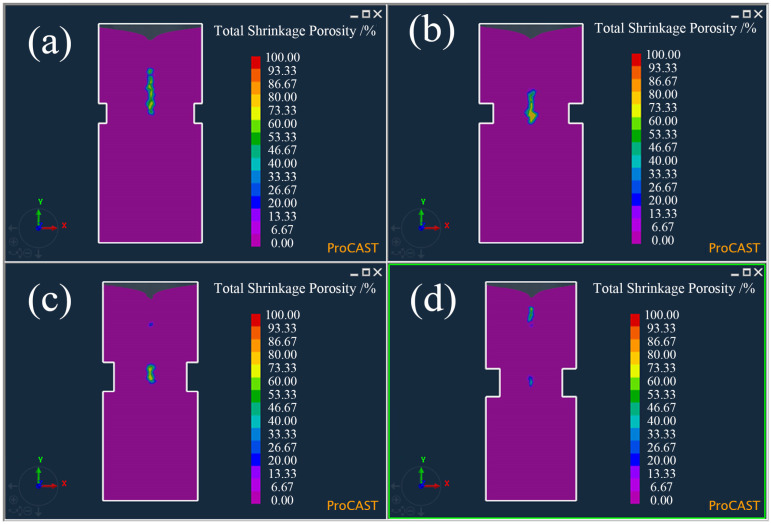
Shrinkage of the Ø26 × 55 mm sample with four different neck sizes by simulation: (**a**) No. 1 in Table 3, (**b**) No. 2 in Table 3, (**c**) No. 3 in Table 3, and (**d**) No. 4 in Table 3.

**Figure 5 materials-15-08413-f005:**
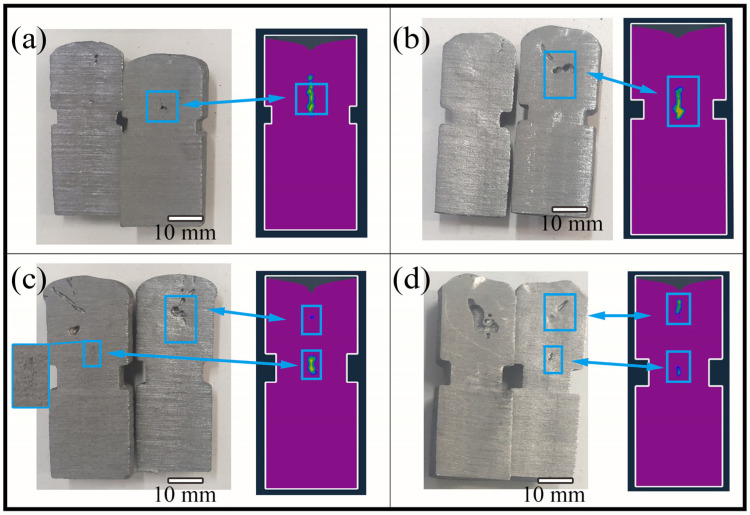
Comparison of experimental and simulation shrinkages of the Ø26 × 55 mm sample with four different neck sizes: (**a**) No. 1 in Table 3, (**b**) No. 2 in Table 3, (**c**) No. 3 in Table 3, and (**d**) No. 4 in Table 3.

**Figure 6 materials-15-08413-f006:**
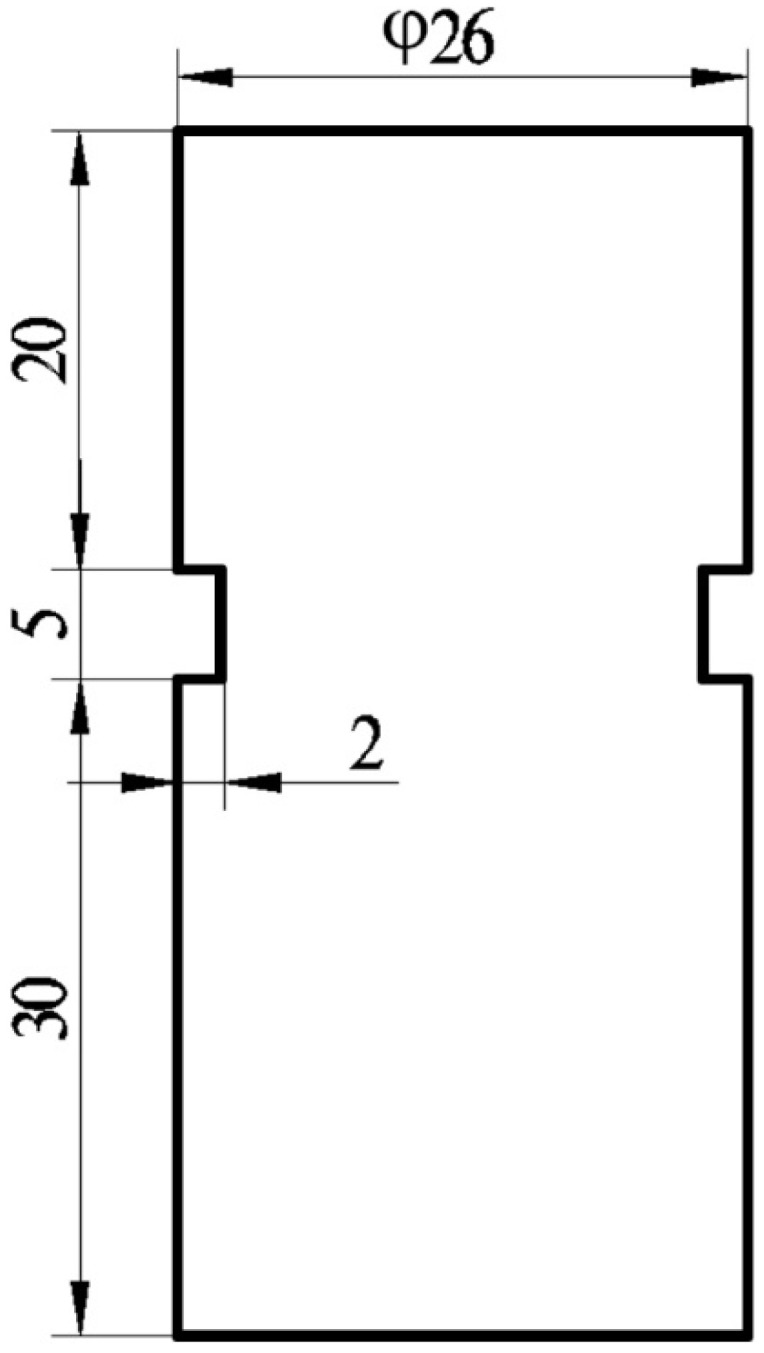
Dimension of the final cylindrical necking test sample (mm).

**Figure 7 materials-15-08413-f007:**
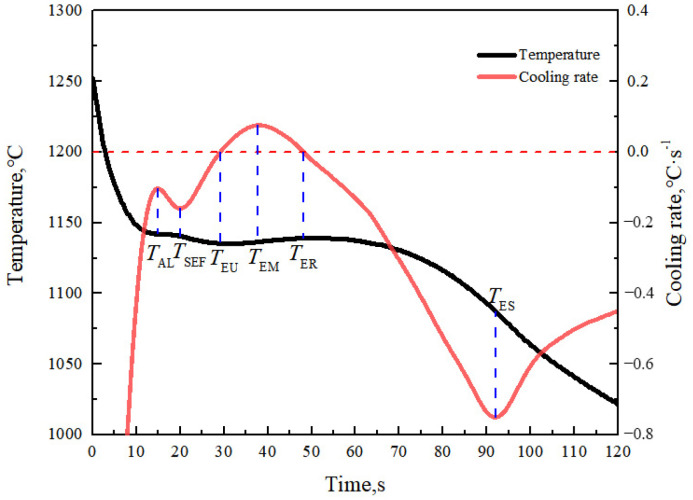
Typical cooling curve, cooling rate curve, and characteristic points of compacted graphite iron.

**Figure 8 materials-15-08413-f008:**
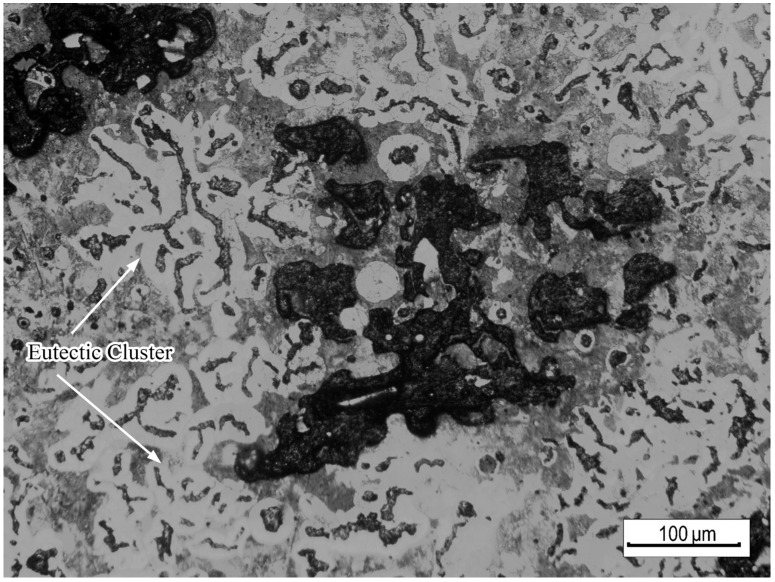
Optical metallographic microstructure around the shrinkage in compacted graphite iron.

**Figure 9 materials-15-08413-f009:**
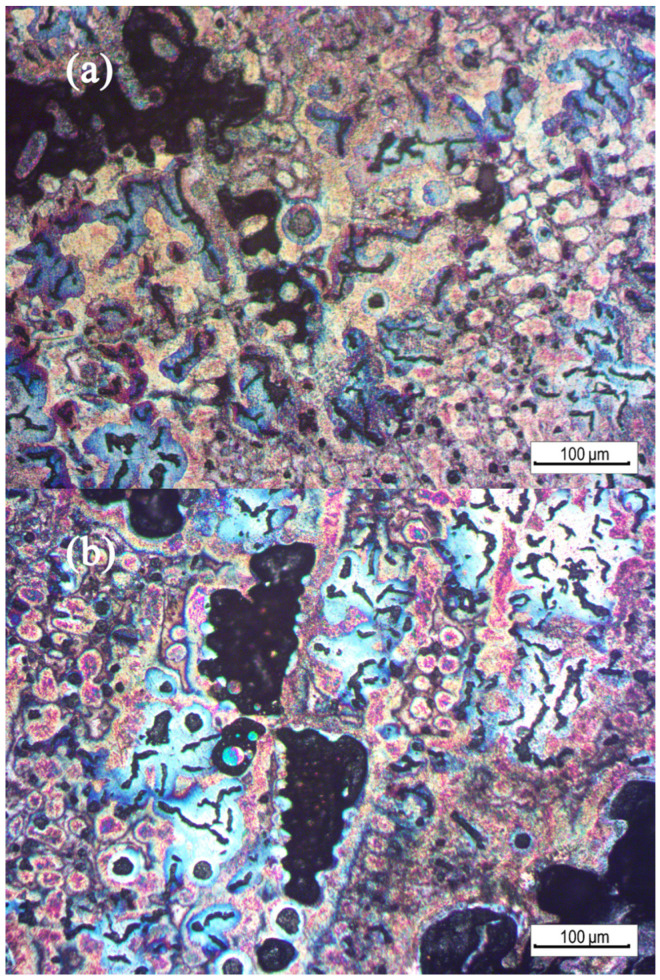
Color metallographic microstructure around the shrinkage: (**a**,**b**) shrinkage in the dendritic gap and (**c**,**d**) shrinkage in the final solidification zone.

**Figure 10 materials-15-08413-f010:**
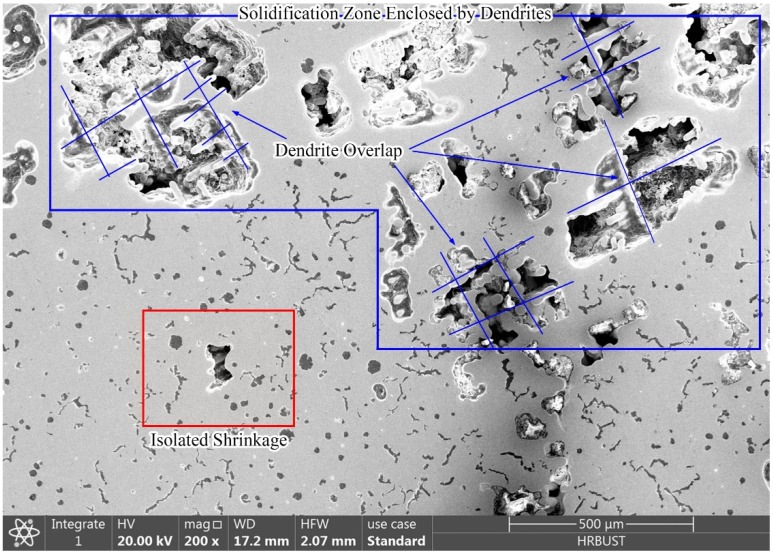
SEM of shrinkage defects in compacted graphite iron.

**Figure 11 materials-15-08413-f011:**
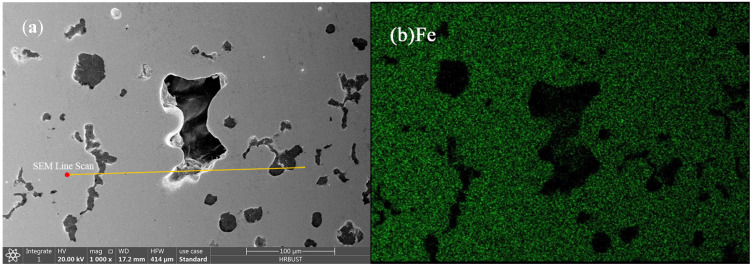
EDS analysis of shrinkage in compacted graphite iron: (**a**) shrinkage morphology and line scan path, (**b**) Fe element distribution, (**c**) C element distribution, (**d**) Si element distribution, and (**e**) line scan curve.

**Figure 12 materials-15-08413-f012:**
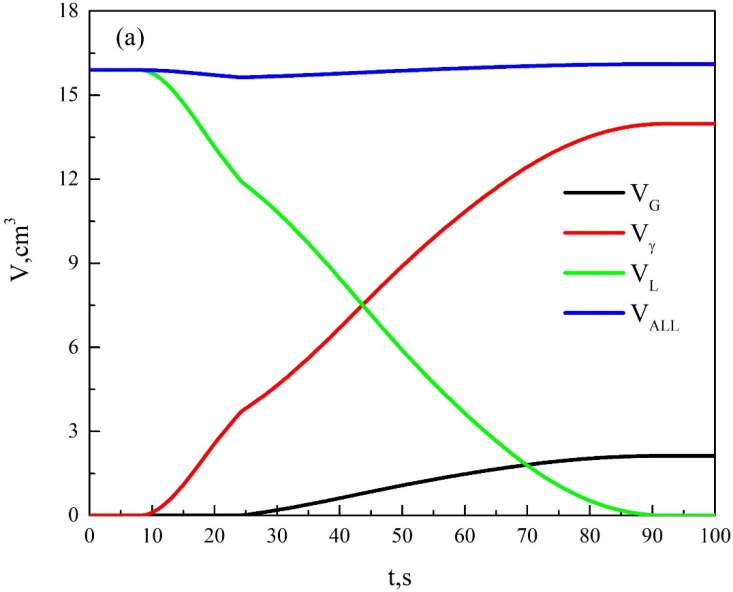
Changes of phase volume and solid-phase rate during solidification: (**a**) change of the phase volume and (**b**) change of the solid phase rate.

**Figure 13 materials-15-08413-f013:**
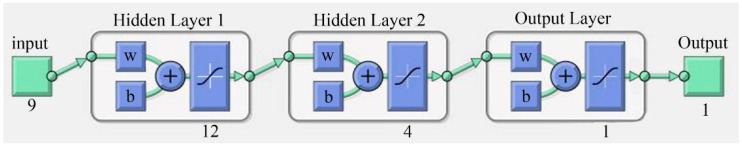
The BP neural network model for predicting the shrinkage tendency of compacted graphite iron.

**Figure 14 materials-15-08413-f014:**
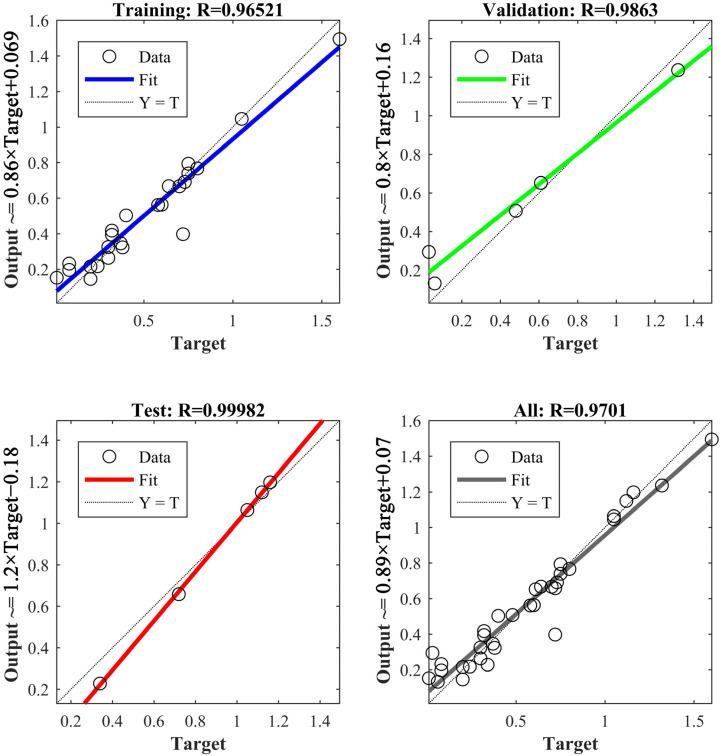
The training and testing results of BP neural network.

**Figure 15 materials-15-08413-f015:**
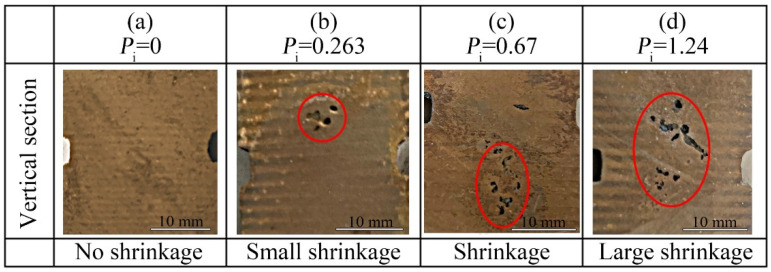
Relationship between *P*_i_ and shrinkage tendency. (**a**) *P*_i_ = 0; (**b**) *P*_i_ = 0.263; (**c**) *P*_i_ = 0.67; (**d**) *P*_i_ = 1.24.

**Table 1 materials-15-08413-t001:** Nominal chemical compositions of compacted graphite iron.

Element	C	Si	Mn	S	P
wt.%	3.7~3.8	2.0~2.4	≤0.6	0.01~0.02	≤0.06
Element	Cu	Mg	Re	Sn	Fe
wt.%	0.3~0.6	0.010~0.018	0.01–0.02	0.04–0.08	Bal.

**Table 2 materials-15-08413-t002:** Four different neck sizes for the Ø26 × 55 mm sample (mm).

No.	Depth of Neck	Height of Neck
1	2	5
2	3	5
3	3	8
4	4	8

**Table 3 materials-15-08413-t003:** Meanings of each thermal analysis characteristic point.

Characteristic Point	Meanings
*T* _AL_	Temperature of austenite formation
*T* _SEF_	Temperature of eutectic freezing
*T* _EU_	The lowest eutectic temperature
*T* _EM_	Temperature of the maximum eutectic reaction rate
*T* _ER_	The highest eutectic temperature
*T* _ES_	Temperature of solidification end

**Table 4 materials-15-08413-t004:** Parameters of a hot alkali corrosion solution and etching process.

wt.%				Etching Process	
NaOH/g	KOH/g	Picric/g	H_2_O/mL	Temperature/°C	Time/min
5	1	0.4	25	95–100	15

**Table 5 materials-15-08413-t005:** Selected thermophysical data.

Parameter	Value
Latent heat of graphite eutectic	$L_{VG}=2028.8\;J/{\mathrm{cm}}^{3}$
Latent heat of austenite	$L_{V\gamma }=1904.4\;J/{\mathrm{cm}}^{3}$
Austenite density	$\rho_{\gamma }=7.51\times 10^{-3}\;\mathrm{kg}/{\mathrm{cm}}^{3}$
Melt density	$\rho_{L}=7.1\times 10^{-3}\;\mathrm{kg}/{\mathrm{cm}}^{3}$
Graphite density	$\rho_{G}=2.3\times 10^{-3}\;\mathrm{kg}/{\mathrm{cm}}^{3}$
Specific heat of melt	$c_{L}=832\;J/\left(\mathrm{kg}\cdot K\right)$
Specific heat of austenite	$c_{\gamma }=649\;J/\left(\mathrm{kg}\cdot K\right)$
Specific heat of graphite	$c_{G}=710\;J/\left(\mathrm{kg}\cdot K\right)$

**Table 6 materials-15-08413-t006:** The meanings and calculation formulas of the thermal analysis characteristic points.

Characteristic Points	Meanings	Formula
$t_{1\left(T_{\mathrm{AL}}-T_{\mathrm{EU}}\right)}$	The time difference between the appearance of *T*_AL_ and *T*_EU_ represents the initial stage of eutectic	$t_{1\left(T_{AL}-T_{EU}\right)}=t_{T_{\mathrm{EU}}}-t_{T_{\mathrm{AL}}}$
$T_{1}$	Temperature difference between *T*_AL_ and *T*_EU_, representing the amount of temperature change at the initial stage of eutectic	$T_{1}=T_{\mathrm{EU}}-T_{\mathrm{AL}}$
$t_{2\left(T_{\mathrm{EU}}-T_{\mathrm{ER}}\right)}$	The time difference between the appearance of *T*_EU_ and *T*_ER_ represents the duration of the eutectic recalescence phase	$t_{2\left(T_{EU}-T_{ER}\right)}=t_{T_{\mathrm{ER}}}-t_{T_{\mathrm{EU}}}$
$T_{2}$	Temperature difference between *T*_EU_ and *T*_ER_, representing the amount of temperature change during the eutectic recalescence phase	$T_{2}=T_{\mathrm{ER}}-T_{\mathrm{EU}}$
$t_{3\left(T_{\mathrm{ER}}-T_{\mathrm{ES}}\right)}$	The time difference between the appearance of *T*_ER_ and *T*_ES_, representing the duration of the end of eutectic	$t_{3\left(T_{ER}-T_{ES}\right)}=t_{T_{\mathrm{ES}}}-t_{T_{\mathrm{ER}}}$
$T_{3}$.	Temperature difference between *T*_ER_ and *T*_ES_, representing the amount of temperature change at the end of eutectic	$T_{3}=T_{\mathrm{ES}}-T_{\mathrm{ER}}$
$t_{4\left(T_{\mathrm{SEF}}-T_{\mathrm{EM}}\right)}$	The time difference between the appearance of *T*_SEF_ and *T*_EM_, representing the duration of the phase of large amounts of graphite generation	$t_{4\left(T_{SEF}-T_{EM}\right)}=t_{T_{\mathrm{EM}}}-t_{T_{\mathrm{SEF}}}$
$T_{4}$	The temperature difference between *T*_SEF_ and *T*_EM_, representing the amount of temperature change in the stage of large amounts of graphite generation	$T_{4}=T_{\mathrm{EM}}-T_{\mathrm{SEF}}$
$Q_{S}$	Definite integral of *T*_SEF_-*T*_EM_ temperature-time curves	$Q_{S}=\int_{t_{T_{\mathrm{SEF}}}}^{t_{T_{\mathrm{EM}}}}f\left(t\right)dt$
$Q_{A}$	Definite integral of *T*_SEF_-*T*_ES_ temperature-time curves	$Q_{Z}=\int_{t_{T_{\mathrm{SEF}}}}^{t_{T_{\mathrm{ES}}}}f\left(t\right)dt$
$D_{t_{\mathrm{TEM}}}$	The value of the derivative at the *T*_EM_	$D_{T_{TEM}}=\underset{\Delta t\to 0}{\lim}\frac{f\left(t_{T_{\mathrm{EM}}}+\Delta t\right)-f\left(t_{T_{\mathrm{EM}}}\right)}{\Delta t}$
$D_{t_{\mathrm{TES}}}$	The value of the derivative at the *T*_ES_	$D_{T_{TES}}=\underset{\Delta t\to 0}{\lim}\frac{f\left(t_{T_{\mathrm{ES}}}+\Delta t\right)-f\left(t_{T_{\mathrm{ES}}}\right)}{\Delta t}$

**Table 7 materials-15-08413-t007:** Standardized canonical correlation coefficient, canonical loadings and cross loadings of each variable.

Variable	Standardized Canonical Correlation Coefficient	Canonical Loadings	Cross Loadings
C	−0.131	0.010	0.007
Si	−0.412	−0.165	−0.123
Mn	0.334	0.436	0.325
P	0.328	0.373	0.278
S	0.144	−0.307	−0.229
Cu	−0.549	−0.316	−0.236
$t_{1\left(T_{\mathrm{AL}}-T_{\mathrm{EU}}\right)}$	−1.008	0.195	0.145
$T_{1}$	−1.988	−0.156	−0.116
$t_{2\left(T_{\mathrm{EU}}-T_{\mathrm{ER}}\right)}$	−0.816	0.022	0.016
$T_{2}$	−1.946	0.025	0.018
$t_{3\left(T_{\mathrm{ER}}-T_{\mathrm{ES}}\right)}$	−0.120	0.209	0.156
$T_{3}$	0.450	0.115	0.086
$t_{4\left(T_{\mathrm{SEF}}-T_{\mathrm{EM}}\right)}$	3.665	0.307	0.229
$T_{4}$	1.292	−0.135	−0.101
$Q_{S}$	−2.532	0.310	0.231
$Q_{Z}$	0.800	0.232	0.173
$D_{T_{TEM}}$	0.932	0.125	0.093
$D_{T_{TES}}$	−0.134	0.058	0.043

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
