# Peer review of "Mechanism of Shrinkage in Compacted Graphite Iron and Prediction of Shrinkage Tendency"

_materials, 2022, doi:10.3390/ma15238413_

Round 1

Reviewer 1 Report

Accept current form of the paper

Author Response

Thank you for your comments concerning our manuscript (ID: materials-2021024). Those comments are all valuable and very helpful for revising and improving our paper. We would like to thank you.

Reviewer 2 Report

This article looks at the mechanism of shrinkage in compacted graphite iron and the tendency toward shrinkage. the obtained results are interesting and this manuscript can be published after revision.

1- abstract : the abstract can be modified the presence of first sentence is not appropriate at the abstract.

2- the introduction section should be changed based on novel articles and extended. there is some references that authors can used them :

Journal of Asian Ceramic Societies 5.4 (2017): 472-478.

Ceramics International 44.9 (2018): 10646-10656.

3) more details should be added to the experimental section

4) could you please explain about the sample preparation for microstructure analysis specially SEM, is there any possibility for presence of carbon due to polishing process i. e. diamond paste, silicon carbide polish plates, etc.

5) it is highly suggested to used dash type curves instead of just changing colors

6) the conclusion should revise based on the most important achievement of this work

Author Response

Dear reviewer and editor,

Thank you for your comments concerning our manuscript (ID: materials-2021024). Those comments are all valuable and very helpful for revising and improving our paper. We would like to thank you and be pleased to answer your questions. The manuscript was revised according to your comments, and all changes were highlighted red color.

The modifications were listed as follows:

This article looks at the mechanism of shrinkage in compacted graphite iron and the tendency toward shrinkage. the obtained results are interesting and this manuscript can be published after revision.

1- abstract : the abstract can be modified the presence of first sentence is not appropriate at the abstract.

The abstract was revised and the expression was replaced.

2- the introduction section should be changed based on novel articles and extended. there is some references that authors can used them :

-  Journal of Asian Ceramic Societies 5.4 (2017): 472-478.

-  Ceramics International 44.9 (2018): 10646-10656.

The introduction was revised and references to the recommended literature were made and were very profitable.

3) more details should be added to the experimental section

Some details have been added to the test section to describe the test materials.

4) could you please explain about the sample preparation for microstructure analysis specially SEM, is there any possibility for presence of carbon due to polishing process i. e. diamond paste, silicon carbide polish plates, etc.

The polishing agent used in the polishing process was Cr2O3, and after polishing, water and anhydrous ethanol were used for cleaning, which did not affect the results of the SEM analysis.

5) it is highly suggested to used dash type curves instead of just changing colors

Figure 12 was modified to use dash curves to indicate the variation of the Solid-phase ratio.

6) the conclusion should revise based on the most important achievement of this work

The conclusions were revised and adjusted, and we thank the reviewer for their comments.

Reviewer 3 Report

The authors have studied the mechanism of shrinkage in compacted graphite iron and prediction of shrinkage tendency by artificial neural network modelling. This is an interesting work and can be useful for steel industries looking for dense cast graphitic steel structures. There are some issues that need to be rectified before publication.

1.      The language is not clear to understand. For example, “Therefore, the cylindrical necking test sample with large modulus is more likely to form gas hole, and it is larger than the shrinkage, which is harmful to the evaluate the 94 shrinkage.”

2.      There is no scale bars present on the microstructure or sample images, Figure 2. Please add proper scale bars.

3.      The sub-captions are not defined in Figure 4. Moreover, the legends and fonts are too tiny to read. Please increase the font size to a suitable value.

4.      Similarly, the sub-captions (a-d) in Figure 5 are not defined. Please improve.

5.      The units of dimensions in Figure 6 are absent. Please add.

6.      Please improve the presentation of Figure 7.

7.      Please add the sub-captions in Figure 9.

8.      What is the basis of selection of data of Table 5?

9.      Additionally English should be improved throughout the manuscript. 

Author Response

Dear reviewer and editor,

Thank you for your comments concerning our manuscript (ID: materials-2021024). Those comments are all valuable and very helpful for revising and improving our paper. We would like to thank you and be pleased to answer your questions. The manuscript was revised according to your comments, and all changes were highlighted red color.

The modifications were listed as follows:

The authors have studied the mechanism of shrinkage in compacted graphite iron and prediction of shrinkage tendency by artificial neural network modelling. This is an interesting work and can be useful for steel industries looking for dense cast graphitic steel structures. There are some issues that need to be rectified before publication.

  1. The language is not clear to understand. For example, “Therefore, the cylindrical necking test sample with large modulus is more likely to form gas hole, and it is larger than the shrinkage, which is harmful to the evaluate the 94 shrinkage.”

The article has been revised and the revised expression reads: Therefore, the cylindrical necking test sample with a large modulus is more likely to form gas hole, The size of the gas holes is larger than the shrinkage, which will result in a large shrinkage rate for the evaluation.

  1. There is no scale bars present on the microstructure or sample images, Figure 2. Please add proper scale bars.

A modification has been made to Figure 2 by adding a scale bar.

  1. The sub-captions are not defined in Figure 4. Moreover, the legends and fonts are too tiny to read. Please increase the font size to a suitable value.

Sub-captions have been added to Figure 3 and Figure 4, and the legend and font have been enlarged.

  1. Similarly, the sub-captions (a-d) in Figure 5 are not defined. Please improve.

Sub-captions and a ruler have been added to Figure 5.

  1. The units of dimensions in Figure 6 are absent. Please add.

The dimension unit: mm is added to the figure caption of Figure 6.

  1. Please improve the presentation of Figure 7.

Improved the wording in the article.

  1. Please add the sub-captions in Figure 9.

Added sub-captions to Figure 9

  1. What is the basis of selection of data of Table 5?

Table 5 was selected based on references 20-22 and has been marked in the article.

  1. Additionally English should be improved throughout the manuscript.

Improvements have been made to the English language throughout the manuscript, and we thank the expert for your valuable comments.

Reviewer 4 Report

In this paper, a cylindrical necking test sample was designed to express the shrinkage tendency in compacted graphite iron based on simulation and experimental verification. The microstructure around the shrinkage zone was observed by hot alkali corrosion method in order to analyze the causes and growth patterns of shrinkage. The generation of each phase during solidification was calculated according to thermodynamic and solidification processes. The shrinkage tendency of compacted graphite iron was predicted by using an artificial neural network approach according to thermal analysis eigenvalues and chemical compositions.

The significant shortcoming and missing of the paper are the following:

1. English should be improved. For example, In ABSTRACT, the semicolons must be replaced by commas. (Line 268): What is it “espresso”? There are also a set of incorrectly constructed sentences into text.

2. Line 70: What is the “BP”?

3. Lines 83-84 and in the subsequent text: F must be replaced by Ø.

4. How numerical modeling was carried out and numerical results were obtained in Figs. 3 – 5?

5. j must be replaced by Ø in Fig. 6.

6. References for Formulae (7) and (8) must be presented. Does undercooling DT depend on time t in (8)?

7. Formula (10): Physical units are not the same in (– TtcLDT).

8. Substituting (13) into (11), formula (14) cannot be obtained.

9. Substituting (15) into (11), formula (16) cannot be obtained.

10. Formula (17): How the ratio of heat changes of phases is coupled with the mass fractions of the phases?

11. Lines 373-376: What the correlation analysis formulae and variables were used? (References?)

12. All references for papers must be added by doi.

Author Response

Dear reviewer and editor,

Thank you for your comments concerning our manuscript (ID: materials-2021024). Those comments are all valuable and very helpful for revising and improving our paper. We would like to thank you and be pleased to answer your questions. The manuscript was revised according to your comments, and all changes were highlighted red color.

The modifications were listed as follows:

In this paper, a cylindrical necking test sample was designed to express the shrinkage tendency in compacted graphite iron based on simulation and experimental verification. The microstructure around the shrinkage zone was observed by hot alkali corrosion method in order to analyze the causes and growth patterns of shrinkage. The generation of each phase during solidification was calculated according to thermodynamic and solidification processes. The shrinkage tendency of compacted graphite iron was predicted by using an artificial neural network approach according to thermal analysis eigenvalues and chemical compositions.

The significant shortcoming and missing of the paper are the following:

  1. English should be improved. For example, In ABSTRACT, the semicolons must be replaced by commas. (Line 268): What is it “espresso”? There are also a set of incorrectly constructed sentences into text.

The semicolon in the abstract has been replaced with a comma, incorrect words have been corrected, and the language of the article has been revised.

  1. Line 70: What is the “BP”?

Explanation of BP: BP (Back Propagation)

  1. Lines 83-84 and in the subsequent text: F must be replaced by Ø.

Changed the symbols in the article.

  1. How numerical modeling was carried out and numerical results were obtained in Figs. 3 – 5?

SolidWorks was used for model building, ProCAST was used for meshing and casting simulation, and finally numerical results were obtained.

  1. j must be replaced by Ø in Fig. 6.

Changed the symbols in the figure.

  1. References for Formulae (7) and (8) must be presented. Does undercooling DT depend on time t in (8)?

A reference to Equations (7) and (8) is provided, while in Equation(8), the undercooling DT is the value on the experimentally determined temperature-time curve corresponding to time t.

  1. Formula (10): Physical units are not the same in (– Tt – cLDT).

Equation 10 has been modified.

  1. Substituting (13) into (11), formula (14) cannot be obtained.

According to the assumptions in the article, equation (14) is obtained by assuming that in term .

  1. Substituting (15) into (11), formula (16) cannot be obtained.

According to the assumptions in the article, equation (16) is obtained by assuming that in term .

  1. Formula (17): How the ratio of heat changes of phases is coupled with the mass fractions of the phases?

The two hypotheses made in the article indicate the heat given off by all austenite production and all graphite production, so the actual heat given off should be in between the two cases, and the volume ratio can be related by the percentage of the actual heat given off in the heat interval of the two hypotheses.

  1. Lines 373-376: What the correlation analysis formulae and variables were used? (References?)

Added references to the typical correlation analysis formulas used, using variables obtained from tests in the experiment such as main elements of iron, the characteristic values of thermal analysis, the initial heat release, and the overall heat release,

  1. All references for papers must be added by doi.

The references were revised and the DOI of the references was added, we thank the expert for your valuable comments.

Round 2

Reviewer 2 Report

IT IS ACCEPTABLE

Author Response

Dear reviewer and editor,

Thank you for your comments concerning our manuscript (ID: materials-2021024). Those comments are all valuable and very helpful for revising and improving our paper. We would like to thank you.

Best wishes,

Zeyu Liu.

Reviewer 4 Report

The authors have introduced significant corrections into text, but some errors remain.

1. See my Point 6 in first review. (Line 294). The author's statement in response: “... in Equation (8), the undercooling DT is the value on the experimentally determined temperature-time curve corresponding to time t.” must be introduced into paper text.

2. See my Points 8 and 9 in first review. According to formula (13), third term in the numerator of formula (14) must be with sign of minus, but not plus. According to formula (15), third term in the numerator of formula (16) must be with sign of minus, but not plus. Did you use error formulae (14) and (6) in your calculations?

3. See my Point 11 in first review. Reader must not search used important formulae in presented references. Therefore, these formulae must be presented into text with corresponding references. In this paper, it relates to the formulae for standardized canonical correlation coefficient, canonical loadings and cross loadings (see Table 7).

4. See my Point 12 in first review. Doi have been added to new references, but all old papers remained without doi.

Author Response

Dear reviewer and editor,

Thank you for your comments concerning our manuscript (ID: materials-2021024). Those comments are all valuable and very helpful for revising and improving our paper. We would like to thank you and be pleased to answer your questions. The manuscript was revised according to your comments, and all changes were highlighted red color.

The modifications were listed as follows:

The authors have introduced significant corrections into text, but some errors remain.

  1. See my Point 6 in first review. (Line 294). The author's statement in response: “... in Equation (8), the undercooling DT is the value on the experimentally determined temperature-time curve corresponding to time t.” must be introduced into paper text.

Changes have been made in the article and these explanations have been introduced into paper text.

  1. See my Points 8 and 9 in first review. According to formula (13), third term in the numerator of formula (14) must be with sign of minus, but not plus. According to formula (15), third term in the numerator of formula (16) must be with sign of minus, but not plus. Did you use error formulae (14) and (6) in your calculations?

Changes have been made to Eqs. (14) and (16), and these two errors are clerical errors that have no effect on subsequent formulas.

  1. See my Point 11 in first review. Reader must not search used important formulae in presented references. Therefore, these formulae must be presented into text with corresponding references. In this paper, it relates to the formulae for standardized canonical correlation coefficient, canonical loadings and cross loadings (see Table 7).

In the typical correlation analysis used in more formulas, given in the text need to take up some meaningless space, this paper uses the typical correlation analysis function of SPSS software for typical correlation analysis of each group of data, to obtain the data shown in Table 7, has been modified for the expression in the text.

  1. See my Point 12 in first review. Doi have been added to new references, but all old papers remained without doi.

Doi has been added to old papers, and we thank the reviewers for their comments.
